# Temperature-Independent Sensor of the Magnetic Field Based on FBG and Terfenol-D

**DOI:** 10.3390/nano13142109

**Published:** 2023-07-19

**Authors:** Shaowei Ma, Haoyu Wu, Shuxian Gao, Meng Sun, Hongyu Song, Qi Wang

**Affiliations:** College of Sciences, Northeastern University, Shenyang 110819, China; mashaowei@stumail.neu.edu.cn (S.M.); wuhaoyu@stumail.neu.edu.cn (H.W.); gaoshuxian@stumail.neu.edu.cn (S.G.); sunmeng@stumail.neu.edu.cn (M.S.); songhongyu@stumail.neu.edu.cn (H.S.)

**Keywords:** magnetic field sensor, Terfenol-D, fiber Bragg grating, temperature compensation

## Abstract

Sensors based on Fiber Bragg Grating (FBG) have remarkable benefits like small size, fast response, wide sensing distribution, and immunity to electromagnetic interference, allowing for their widespread application in numerous domains of physical parameter measurement in industrial engineering. In this work, a temperature-independent sensor of the magnetic field based on FBG and the magnetostrictive material Terfenol-D is suggested. By exploiting the distributed sensing characteristic of FBG, a sensing structure that remains unaffected by temperature is designed. The results demonstrate that within the magnetic induction intensity range of 0 mT to 50 mT, the sensitivity of the sensor can reach 7.382 pm/mT, exhibiting good linearity and repeatability. Compared with the control experiment and other sensors of the magnetic field containing Terfenol-D, the sensor has higher sensitivity, better repeatability, and good temperature stability.

## 1. Introduction

Currently, sensors of the magnetic field are widely used in various fields, such as fault diagnosis of industrial machinery [1,2], biomedical detection [3,4], electronic navigation monitoring [5], aerospace [6,7], and more. Conventional sensors of the magnetic field in use today are based on principles such as the Hall effect [8,9], the Faraday law of electromagnetic induction [10], the Josephson effect [11], the Meissner effect [12], the Magnetoresistance effects [13,14], and the Giant Magnetoresistance effect [15,16]. However, these sensors have a complex structure, high cost, and are susceptible to electromagnetic interference to some degree. With the advancement of technology, fiber-optic-based sensors of the magnetic field [17,18] are becoming increasingly popular because of their smaller size, lower cost, resistance to electromagnetic interference, and better sensitivity.

There are three types of sensors of the magnetic field based on optical fiber that are now in use. The first type is fiber optic interferometric sensors, such as Mach–Zehnder interferometers [19,20], Fabry–Perot interferometers [21], Sagnac interferometers [22], and Michelson interferometers [23]. The principle of these sensors is to sense the physical parameter changes in the environment by detecting the relative phase shift variation between two optical pathways. According to the change of interference effect of two optical paths, the relevant physical parameters can be sensed. Most of these sensors achieve magnetic field sensing by combining with magnetic fluid [19,21,24,25] and using the magnetic fluid’s magneto-optical effect to exchange energy. However, due to the limited changeable range of the magnetic fluid’s refractive index, the sensor’s detecting range is limited. The second type is sensors based on evanescent fields, which mostly use tapered fibers [26,27], D-shaped fibers [28,29,30], and U-shaped fibers [31,32]. The principle of these sensors is to expose the core area of the optical fiber as much as possible by processing it so that the light propagating in the fiber is exposed from the exposed area to form an evanescent field, which interacts with the environment to sense the physical parameters. Sensors of the magnetic field based on this principle can not only be combined with magnetic fluids [26,29,32,33] like the first category but can also be implemented by coating the exposed core area with a film and utilizing the surface plasmon resonance (SPR) effect [30,34,35] to sense the magnetic field. The fiber structure required by these sensors is complex and difficult to process and is easily affected by other external environmental factors besides the measured physical parameters.

The third type of sensor is based on fiber optic gratings, which mostly use fiber Bragg gratings [36,37], long-period fiber gratings [38,39], and tilted fiber gratings [40] for sensing structures. Fiber optic gratings can sense physical parameters by measuring changes in their wavelength parameters. The refractive index of the fiber core varies periodically along the fiber’s axial direction, forming a narrow-band filter or reflector. In magnetic field sensing, apart from combining with magnetic fluid [37,41] and coating to achieve the SPR effect [30,42], fiber Bragg gratings (FBGs) can also be combined with magnetostrictive materials [43,44,45,46,47]. In the presence of a magnetic field, the magnetic domain in magnetostrictive materials will rotate and suffer directional deformation due to the rotation of the magnetic domain [48]. When FBGs are subjected to environmental conditions like strain and temperature, their central wavelength shifts, which can be utilized to detect the deformation of magnetostrictive material [49] and achieve magnetic field sensing. Sensors with this structure are simple in design and immune to electromagnetic interference.

As giant magnetostrictive materials (GMMs), Terfenol-D has been widely applied in sensor structures combined with FBGs. In 2016, H. García-Miquel et al. presented a sensor which was based on the rod-shaped Terfenol-D GMMs and FBGs for strain and temperature monitoring [50]. Terfenol-D magnetoelastic characteristics are affected by magnetization intensity, applied force, and temperature. The sensor designed a sensing network consisting of four FBGs, where two perpendicular FBGs measured the transverse and longitudinal magnetostriction of the Terfenol-D, when the other two FBGs measured temperature and strain and amplified the strain via a mechanical system, improving the sensor’s performance. In 2022, Claudio Floridia et al. designed a structure based on Terfenol-D and FBG to simultaneously measure current and temperature [51]. This study designed an encapsulation structure, in which Terfenol-D and fiber gratings were glued together and put in the magnetic field that was generated by the current. The center wavelength shift of FBG was measured for changes in current and temperature, and the surface curve graph of the two sets of measurement results was plotted. The intersection point in the graph was used to simultaneously sense the current and temperature.

These studies demonstrate that the elimination of temperature effects on FBG-based sensors and the realization of single physical parameter measurements are important issues worthy of research. To solve this problem, Yanxiao He proposed a PZT-FBG voltage sensor in 2021 that is not affected by temperature changes. This sensor combines piezoelectric ceramics (PZT) with FBG [52]. The PZT converts the applied voltage on the material into deformation, which is then monitored by FBG to detect the voltage. The sensor uses a double FBG structure and senses the measured voltage by measuring the peak area of the output spectrum. This structure effectively eliminates the influence of temperature on voltage sensing.

In this work, a temperature-independent sensor of the magnetic field based on fiber Bragg grating and Terfenol-D is proposed. First, the basic principles of Terfenol-D’s magnetostrictive effect are analyzed to determine the direction and amount of strain caused by a magnetic field, which leads to elongation or contraction of the fiber along its axis. Single FBG and Terfenol-D sensor of the magnetic field are combined and its sensitivity is calibrated. Subsequently, a temperature-independent sensor of the magnetic field is designed by combining Terfenol-D with double FBGs, and its sensitivity is compared with that of the single FBG and Terfenol-D combination. Finally, the temperature stability of the two sensors is examined. The results demonstrate that the double FBG and Terfenol-D sensor has excellent temperature stability and higher sensitivity compared to the sensors described in previous articles.

## 2. Sensor Principles

In this section, the principle of device sensing is analyzed. The properties of Terfenol-D are analyzed and characterized. According to the FBG sensing principle, the basic structure of the sensor which is temperature-independent is designed.

### 2.1. Characteristic of Terfenol-D

Terbium (Tb) is an important rare earth element, and Terfenol-D, a giant magnetostrictive material (GMMs), is an alloy of Tb. Terfenol-D has a cubic crystal structure, and its magnetostriction coefficient is generally obtained by the length change along the principal axis (110) [53]. When a magnetic field is placed along its principal axis, the material produces deformation parallel to the principal axis direction. Terfenol-D saturated magnetostrictive strain is large, energy density is high, magneto-mechanical coupling coefficient is large, sound velocity is low, and Curie temperature is high. In addition, compared with piezoceramics that will completely disappear permanently due to overheating, Terfenol-D will only temporarily lose its magnetostrictive characteristics when operating above the Curie temperature, and its magnetostrictive characteristics can be completely restored when cooling below the Curie temperature, so there is no overheating failure problem [53]. The magnetostriction size is represented by the magnetostriction coefficient λ = ∆D/L_m_, where ∆D is the material length change and Lm is the total length of the material, as illustrated in Figure 1a. The principle of magnetostriction is shown in Figure 1b. Microscopically, magnetic materials can be regarded as having many magnetic domains. The magnetic moments of these domains have distinct orientations when the external magnetic induction intensity is zero, and the vector total is zero. Therefore, the material has no magnetization and no deformation on a macroscopic level [54]. When these microscopic magnetic domains are exposed to an external magnetic induction intensity, they will rotate from their original self-magnetization direction and turn towards the external magnetic induction intensity direction. While the magnetic domains have a single orientation, they will turn in a single direction and produce magnetic induction intensity force in that direction, causing the magnetostrictive material to experience force in this direction, thus producing deformation, known as magnetostriction on a macroscopic level. Within a specific range, the magnetostriction of the material will increase with increasing external magnetic induction intensity and have good linearity. However, when the applied magnetic induction intensity is beyond this range, its linearity will decrease until it no longer deforms, and the material deformation will reach saturation.

Table 1 presents some property parameters of Terfenol-D at 25 °C. It can be observed that Terfenol-D, as a magnetostrictive material, has a higher saturation magnetostriction coefficient than other conventional magnetostrictive materials at room temperature [55] and also has certain thermal expansibility. From these parameters, it can be inferred that the magnetostriction of Terfenol-D is influenced by magnetic induction intensity and temperature. Figure 2 depicts the connection between its magnetostrictive ratio and magnetic induction intensity. The data in the figure were obtained through standard strain gauge testing at 5 Mpa and room temperature (25 °C) using a (110) single-oriented Terfenol-D sensing material. It can be seen that the magnetostriction of this material has good linearity in the low magnetic induction intensity range. The Terfenol-D sensing material and related parameters, as well as its magnetostriction testing data, are provided by Shijiazhuang Saining Electronic Technology Co., Ltd. (Shijiazhuang, China).

The diffraction spectrum of Terfenol-D is analyzed using X-ray diffraction for qualitative analysis of its composition. As Terfenol-D is an alloy of specific proportions of Tb, Dy, and Fe, its XRD spectrum was compared to the standard card XRD spectra of DyFe2 (PDF #25-1094) and Fe2Tb (PDF #33-0680) in Figure 3. The diffraction peaks appeared near 2θ = 20.984°, 34.535°, 40.739°, 42.632°, 55.990°, 61.889°, 66.068°, 72.868°, 83.129°, 87.002°, and 88.216°, corresponding to the crystal planes (111), (022), (113), (222), (024), (224), (333), (044), (026), (335), and (226), which can be matched with the two standard cards, indicating that the material is an alloy of specific proportions of Tb, Dy, and Fe. According to the XRD pattern, the average crystallite size is calculated using the Scherrer equation. The average crystallite size of Terfenol-D used is 20.410 nm. At this size, the strength and hardness of the alloy material are lower, but the larger the grain, the less the grain interface, the less the grain boundary interleaving, the better the magnetostriction performance. The sharp shape of each peak reflects the integrity of the material. X-ray diffraction is performed using Bruker D8 Advance from Japan.

As shown in Figure 4, the surface microstructure of sheet Terfenol-D is collected by scanning electron microscope (SEM). The red line in the figure indicates the magneto striction direction customized by the material and the orientation of the crystal principal axis (110). When the magnetic field is supplied along principal axis direction, the material deforms parallel to the direction of the spindle, resulting in magnetostriction phenomenon. Scanning electron microscopy is performed using Zeiss Sigma 300 from Japan.

### 2.2. Sensor Mechanism

The FBG serves as a wavelength filter, utilizing a fiber grating that modulates refractive index periodically along its axis. Within the fiber core, Bragg grating acts as a reflecting mirror which reflects light that satisfies the Bragg grating wavelength condition of the reflected light, while transmitting light of other wavelengths. The center wavelength of the reflection spectrum is the Bragg wavelength λ_B_, which is influenced by the grating period Λ and the effective refractive index n_eff_ of the fiber core. The reflected spectrum wavelength can be given as follows:(1)$$\lambda_{B}=2n_{eff}\Lambda$$

When the FBG is subjected to temperature or strain, its grating period Λ or effective refractive index n_eff_ changes, causing shifts in the center wavelength. The Bragg center wavelength shift ∆λ_B_ is explained below [56],

(2)$$∆\lambda_{B}=\lambda_{B}\left(1-\rho \right)\frac{∆L_{\mathrm{FBG}}}{L_{\mathrm{FBG}}}+\lambda_{B}\left(\alpha +\xi \right)∆T$$
where ρ is the effective strain-optic coefficient; L_FBG_ is the length of the grating; ∆L_FBG_ is the change in the grating length; α is the thermal expansion coefficient; ξ is the thermo-optic coefficient; and ∆T is the temperature variation.

Figure 5 shows that the sensing principle involves the conversion of an externally applied magnetic field into Terfenol-D strain ∆D through its magnetostriction effect.

FBG and Terfenol-D are directly bonded using optical glue, whereby Terfenol-D strain ∆D is transformed into FBG length variation ∆L_FBG_. FBG length variation is equal to the Terfenol-D strain. The equal relation of the two variables is as follows:(3)$$∆L_{FBG}=∆D$$

If the effect of temperature in the environment can be ignored, according to Figure 2, Terfenol-D deformation demonstrates remarkable linearity with magnetic induction intensity in the low field region. Combining Equation (2), ∆λ_B_ can be described as follows,
(4)$$∆\lambda_{B}\propto \lambda_{B}\left(1-\rho \right)\frac{∆B}{L_{FBG}}$$

Hence, the magnetic induction intensity can be obtained directly from the variation of the central wavelength.

### 2.3. Sensor Basic Configuration

Figure 5 depicts the fundamental sensing unit. Terfenol-D converts the external magnetic induction intensity signal into deformation along its principal axis (110) (*z*-axis). The fiber Bragg grating (FBG) is directly bonded to Terfenol-D using optical glue. By utilizing the center wavelength drift characteristic of FBG, the deformation signal is converted to the center wavelength shift.

The shift of central wavelength of the FBG reflection spectrum of the temperature-independent sensor is shown in Equation (4). Figure 6a is the basic layout of the sensor, where the sensing and reference units are composed of the identical structure of FBG and Terfenol-D. Two FBGs have similar center wavelengths. The input broadband spectrum first enters the sensing unit, and the spectral band of the reflected light of FBG1 enters FBG2 through the optical circulator. At this time, only the overlapping area of the reflection spectra of FBG1 and FBG2 will be reflected by the optical circulator, which becomes the final output spectrum of the entire sensor system. According to Figure 6b, when the sensing unit receives just the magnetic field application, without changing the external environmental temperature, the reflection spectrum of FBG1 will shift according to the variation of magnetic induction intensity, while the reflection spectrum of FBG2 does not change. As a result, the peak width of the output spectrum of the sensor system changes. The sensing unit and reference unit of the sensor system uses the same structure of FBG and Terfenol-D, with the same temperature coefficient of FBG1 and FBG2 and the same thermal expansion coefficient of Terfenol-D1 and Terfenol-D2. Therefore, the reflection spectrum shift of FBG1 and FBG2 caused by temperature is identical. When the external temperature changes, the overlapping area of the spectra of the double FBGs in the final output will not change, and the magnetic field sensing is not affected by temperature changes. Compared with the previous double FBG sensor that eliminates temperature influence by measuring the area of the overlapping region of the output spectra [52], the response function of the measured peak width change has better linearity.

## 3. Sensor Performance Test

This part fabricates a sensor prototype in accordance with the ideal design discussed above and evaluates its performance. Sensitivity calibration is carried out for both the single FBG and Terfenol-D coupled sensor of the magnetic field, as well as the temperature-independent double FBG and Terfenol-D coupled sensor of the magnetic field. The temperature stability of these sensors is also tested. Finally, the sensitivity of the proposed sensor is compared with others in the literature.

### 3.1. Sensitivity Calibration

The basic unit of the sensor, as shown in Figure 7a, uses the optical glue to bond the fiber Bragg grating (FBG) and the magnetostrictive material (Terfenol-D). To achieve high reflectivity, the grating region of the FBG used in the experiment has a length of 10 mm, a center wavelength of 1540.94 ± 0.02 nm, and a reflectivity ≥ 85%. A sheet of Terfenol-D (20 mm × 15 mm × 1 mm) is used to ensure complete coverage of the magnetostrictive area of the grating region, leading to sufficient deformation. Optical glue is utilized to secure both ends of the grating region to the Terfenol-D, thereby avoiding the impact of the glue on the grating’s deformation. To simulate real magnetic field environments, the magnetic field is provided by a magnet, and its magnetic induction intensity is measured by a Tesla meter for sensitivity calibration. The relationship between the magnetic induction intensity of the magnet and the distance from the magnet is illustrated in Figure 7b. Input light is provided by Fiber Laser Source (KG-ASE-CL-D-17-PC, Conquer, wavelength range 1528 nm~1603 nm). The output spectrum of the fiber grating is recorded by the Optical Spectrum Analyzer (AQ6370D, Yokogawa, Singapore) with a step size of 5 mT, and the reflective spectra changes within the range of 0 mT to 50 mT magnetic induction intensity are measured.

Figure 8 is the testing setup for the sensor of the magnetic field utilizing the single fiber Bragg grating (FBG) combined with Terfenol-D (control group). The experiment involved three cycles of increasing and decreasing magnetic fields to assess the sensor’s repeatability and hysteresis response to magnetostriction. The changes in the reflectance spectrum and center wavelength of the single FBG and Terfenol-D are illustrated in Figure 9a. The results show that when the magnetic induction intensity increases, the central wavelength of FBG shifts red, and when the magnetic induction intensity decreases, the central wavelength shifts blue. Figure 9b shows the results of the three-cycle repeated experiment, demonstrating good repeatability of the sensor during the process of increasing and decreasing magnetic induction intensity. Figure 9c displays the linear fit of the average response of the sensor obtained from the repeated experiment at 0 mT to 50 mT magnetic induction. The sensor has an average sensitivity of 6.093 pm/mT and exhibits good linearity. The goodness of the linear fit is more than 0.98, which can ensure the detection resolution. As shown in Figure 9c, the sensitivity of the sensor differs during the process of increasing and decreasing magnetic induction intensity, showing a slight hysteresis effect with a hysteresis error of 4.88%, which is in accordance with the magnetostriction delay effect of the Terfenol-D [50] and is acceptable for the experimental repeatability.

The sensor of the magnetic field combined with single FBG and Terfenol-D shows good performance in magnetic field measurement. However, because the central wavelength shift of FBG is affected by temperature, the temperature stability of the sensor is poor, which will affect its magnetic field sensing performance. Therefore, the sensor of the magnetic field combined with double fiber grating and Terfenol-D is designed in this paper, which not only improves the temperature stability, but also improves the sensitivity of the sensor.

Figure 10 is the testing setup for the sensor of the magnetic field that combines the double fiber Bragg grating (FBG) with Terfenol-D, which is not affected by temperature. The experiment involved three cycles of the same process as the control group. The changes in the reflectance spectrum and peak width of the double FBG are shown in Figure 11a. As the magnetic induction intensity increases, the peak width of the spectrum decreases, and as it decreases, the peak width increases. The results of the three-cycle repeated experiment are shown in Figure 11b, where the sensor exhibited excellent repeatability during the process of increasing and decreasing the magnetic induction intensity. Figure 11c shows the linear fit of the sensor’s average response obtained from the repeated experiments at 0 mT to 50 mT magnetic induction. The sensor has an average sensitivity of 7.382 pm/mT and exhibits good linearity. The goodness of the linear fit is more than 0.98 which can ensure the detection resolution. As shown in Figure 11c, the sensor’s sensitivity varies slightly during the processes of increasing and decreasing the magnetic induction intensity, showing a slight hysteresis effect with a hysteresis error of 5.89%, which is consistent with the magnetostriction delay effect of the Terfenol-D [50]. This effect’s impact on the experiment’s repeatability is acceptable.

Figure 12 compares the linear fits of the average responses of the single FBG and double FBG sensors combined with Terfenol-D during the process of increasing and decreasing the magnetic induction intensity. The double FBG sensor combined with Terfenol-D is evidently more sensitive than the single FBG sensor combined with Terfenol-D, with a sensitivity increase from 6.093 pm/mT to 7.382 pm/mT. Table 2 presents a comparison of this sensor with other sensors of the magnetic field based on FBG and Terfenol-D [45,46,47].

### 3.2. Temperature Stability Test

The temperature stability test is conducted using the Microcomputer Heating Platform (JF-956C, Jftoois, Valmiera, Latvija) at temperatures ranging from 30 °C to 50 °C, with a heating temperature accuracy of 0.1 °C. The output spectrum of the sensor is recorded using an Optical Spectrum Analyzer (AQ6370D, Yokogawa, Singapore) with a step size of 1 °C, to measure the reflected spectrum changes within the temperature range of 30 °C to 50 °C. Three repeated experiments are performed, and the average responses of the single FBG and Terfenol-D and double FBG and Terfenol-D during temperature changes are shown in Figure 13. It can be seen that the sensor combined with single FBG and Terfenol-D is sensitive to temperature changes, with wavelength offset up to 260 pm, while the sensor combined with double FBG and Terfenol-D has a slightly fluctuating peak width with temperature changes, keeping within the range of 15 pm. The results show that the double FBG and Terfenol-D sensor can achieve temperature compensation and eliminate the thermal spectrum shift caused by the thermal expansion of the alloy material and the effective refractive index n_eff_ change of the fiber core caused by temperature [52,57], resulting in a fully magnetically induced output spectrum change with good temperature stability.

## 4. Conclusions

In this work, a temperature-independent sensor of the magnetic field based on fiber Bragg grating and giant magnetostrictive material Terfenol-D is proposed. The sensor features a simple probe design, compatibility with optical fibers, resistance to electromagnetic interference, good temperature stability, and has great potential for wide application in electric engineering. Through a designed control experiment, the performance of the proposed temperature-independent sensor of the magnetic field is tested. Experimental results show that the sensitivity of the proposed sensor, which is not affected by temperature, can reach 7.382 pm/mT in the range of 0 mT to 50 mT magnetic induction intensity, with good linearity and repeatability. The goodness of the linear fit is more than 0.98 which can ensure the detection resolution. Additionally, temperature stability testing shows that the designed double FBG and Terfenol-D sensor of the magnetic field has good temperature stability compared to a control group experiment without a temperature compensation structure. Therefore, the designed sensor of the magnetic field has excellent performance.

## Figures and Tables

**Figure 1 nanomaterials-13-02109-f001:**
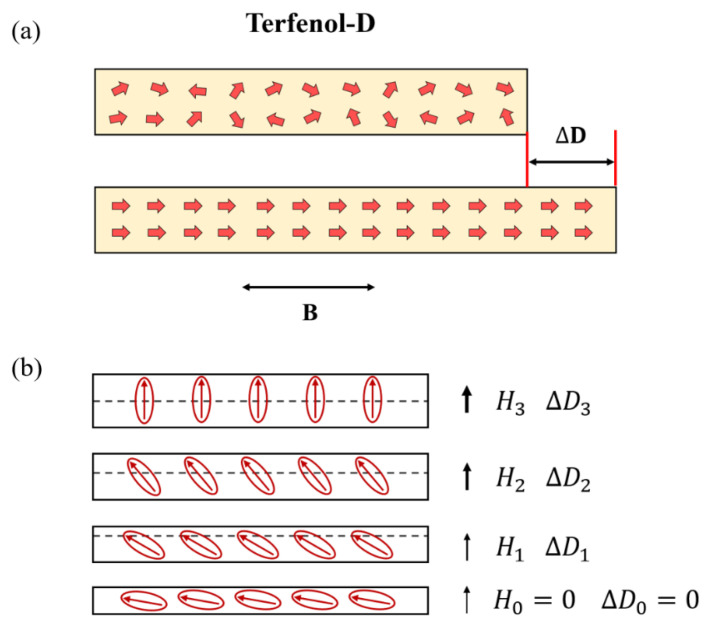
(**a**) Magnetostriction phenomenon, (**b**) Principle of magnetostriction.

**Figure 2 nanomaterials-13-02109-f002:**
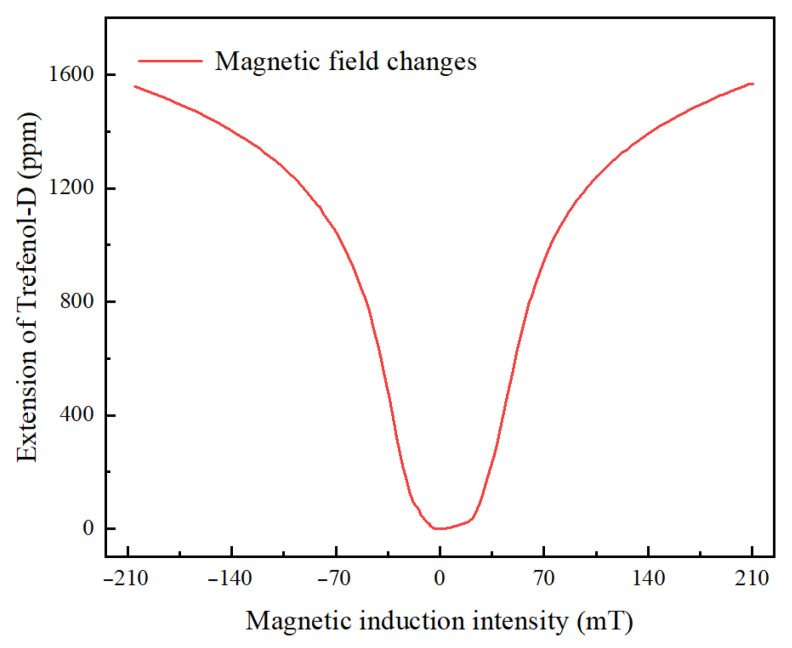
The connection between magnetic induction intensity and extension of Terfenol-D.

**Figure 3 nanomaterials-13-02109-f003:**
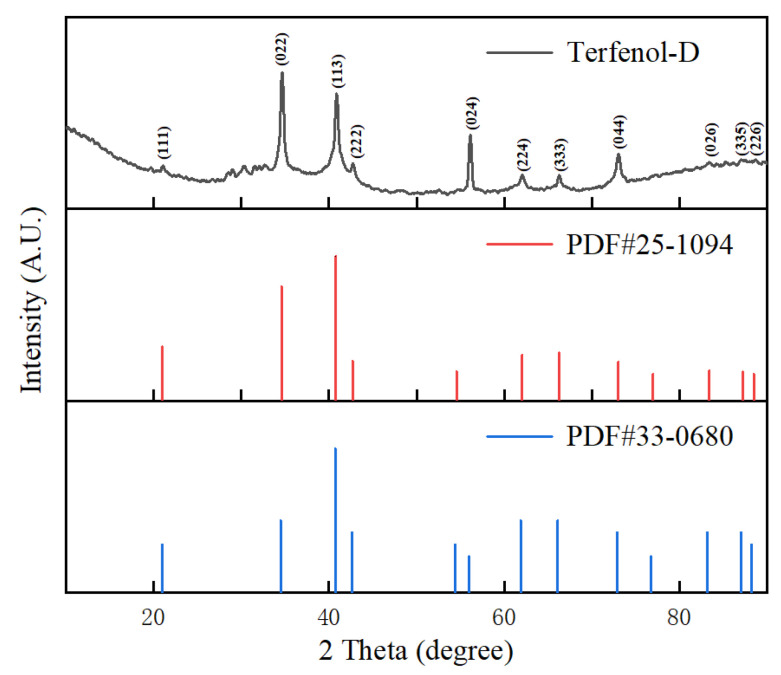
XRD pattern of Terfenol-D.

**Figure 4 nanomaterials-13-02109-f004:**
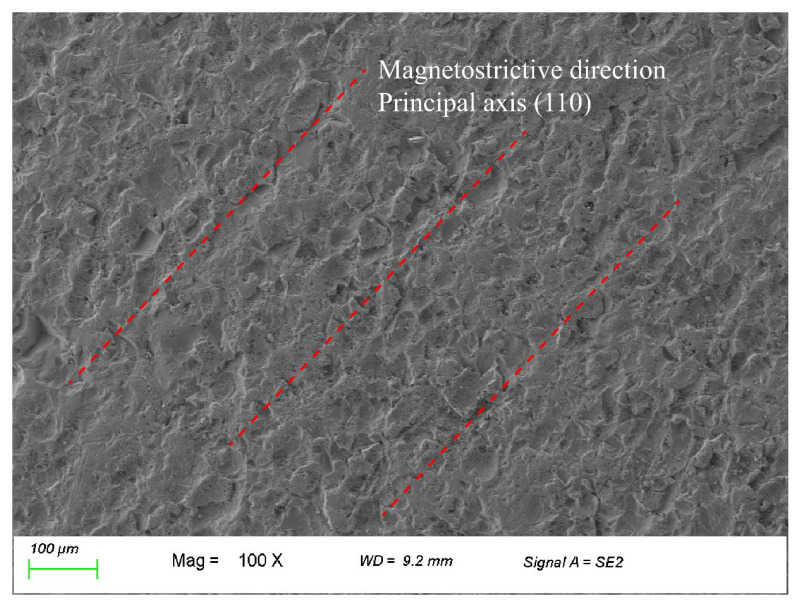
Surface Microstructure of sheet Terfenol-D.

**Figure 5 nanomaterials-13-02109-f005:**
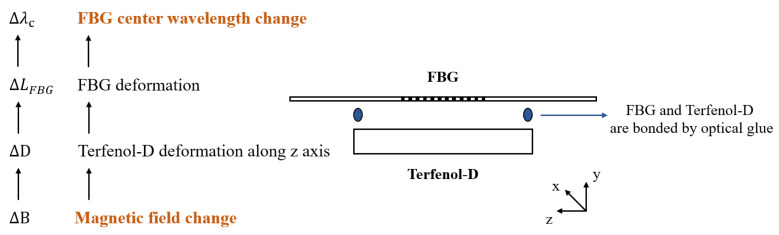
FBG and Terfenol-D sensor of the magnetic field mechanism.

**Figure 6 nanomaterials-13-02109-f006:**
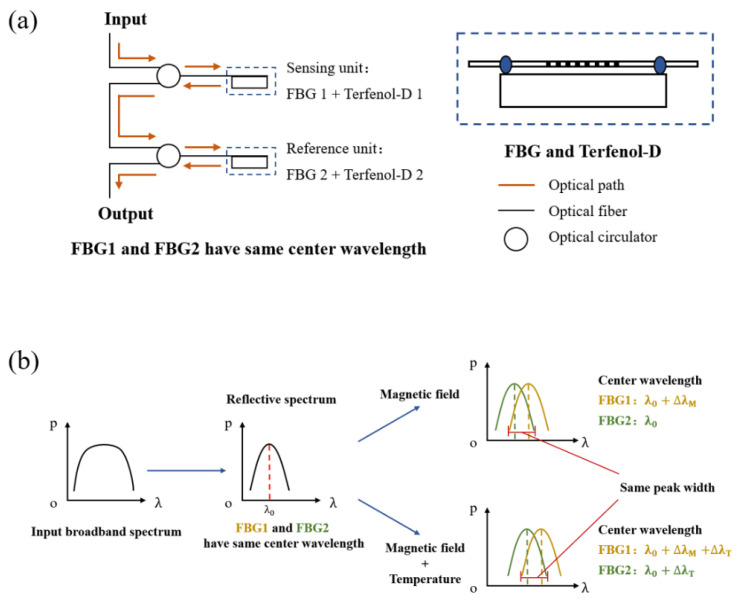
(**a**) Sensor basic configuration, (**b**) Temperature compensation principle of sensor.

**Figure 7 nanomaterials-13-02109-f007:**
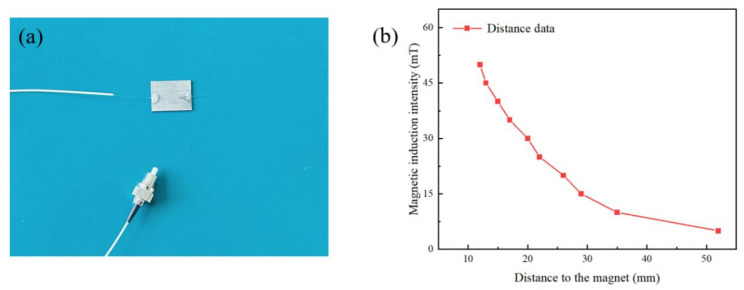
(**a**) Basic unit of the sensor, (**b**) The relationship between the magnetic induction intensity of the magnet and the distance from the magnet.

**Figure 8 nanomaterials-13-02109-f008:**
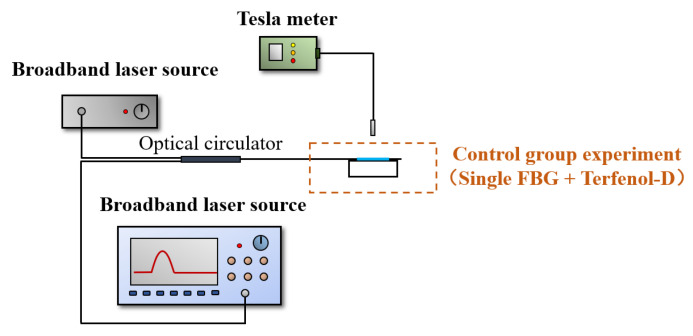
Experimental setup of single FBG and Terfenol-D.

**Figure 9 nanomaterials-13-02109-f009:**
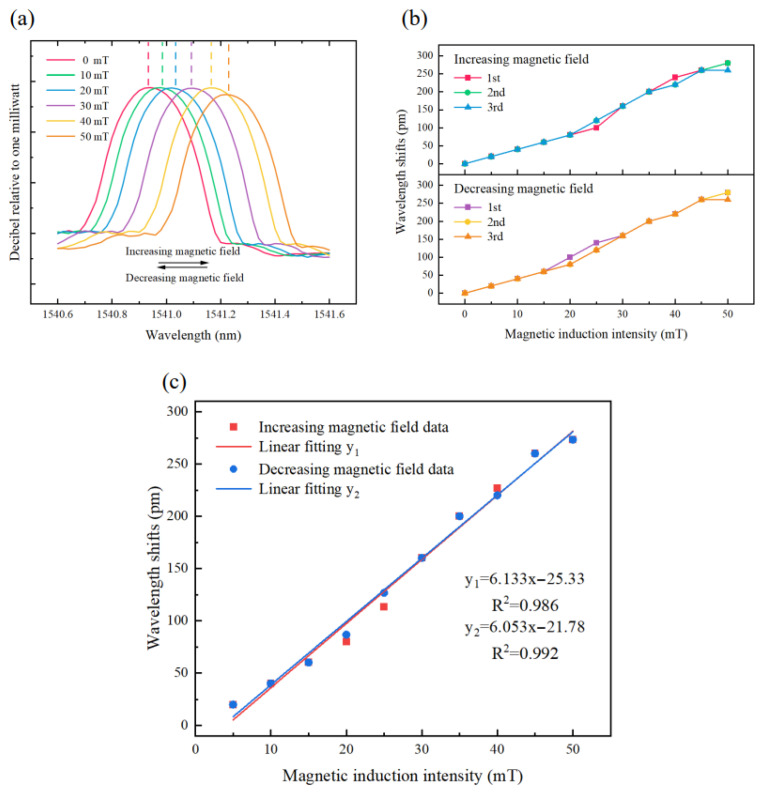
(**a**) The changes in the reflectance spectrum and center wavelength of the single FBG and Terfenol-D, (**b**) Average response of three tests, (**c**) The linear fit of the average response.

**Figure 10 nanomaterials-13-02109-f010:**
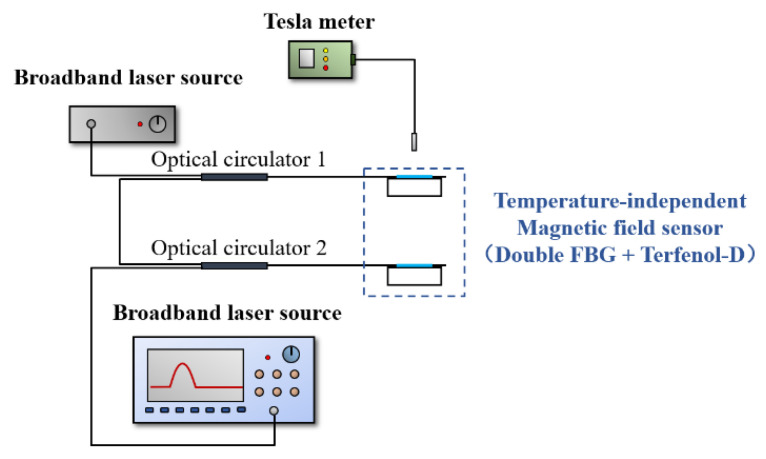
Experimental setup of double FBG and Terfenol-D.

**Figure 11 nanomaterials-13-02109-f011:**
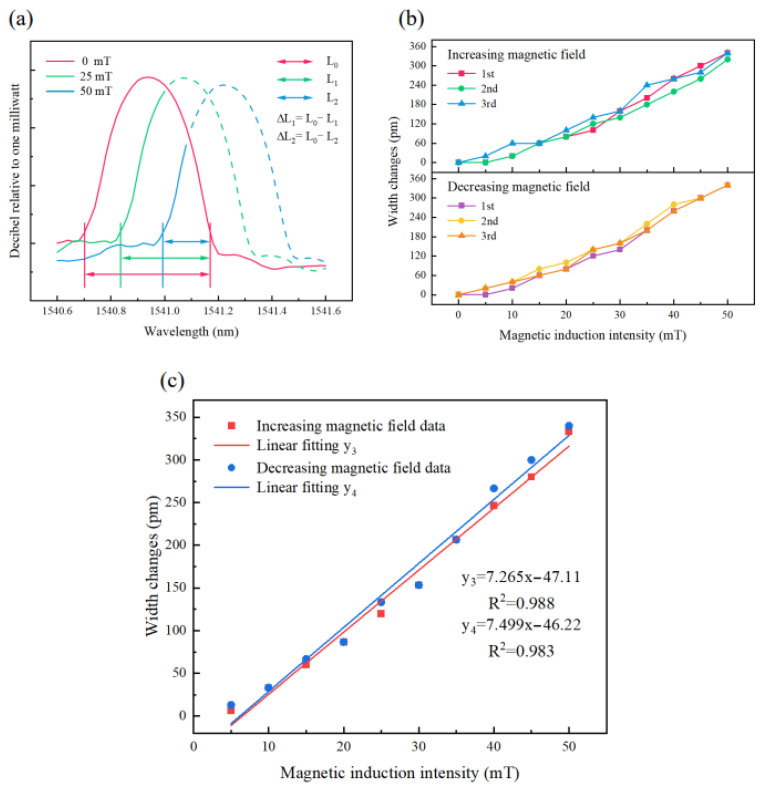
(**a**) The changes in the reflectance spectrum and center wavelength of the double FBG and Terfenol-D, (**b**) Average response of three tests, (**c**) The linear fit of the average response.

**Figure 12 nanomaterials-13-02109-f012:**
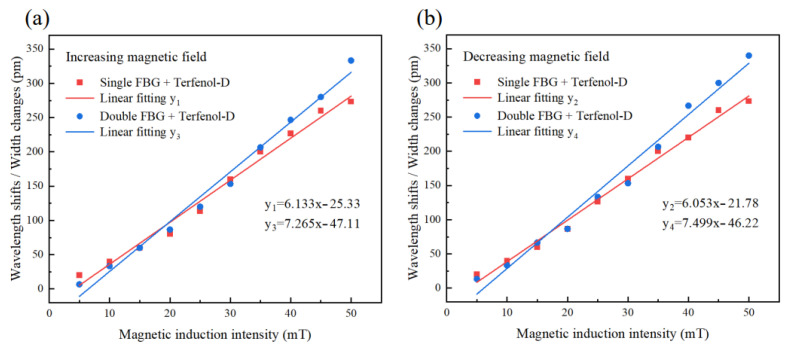
Sensor sensitivity of the single FBG and Terfenol-D and the double FBG and Terfenol-D, (**a**) Magnetic induction intensity increasing process, (**b**) Magnetic induction intensity decreasing process.

**Figure 13 nanomaterials-13-02109-f013:**
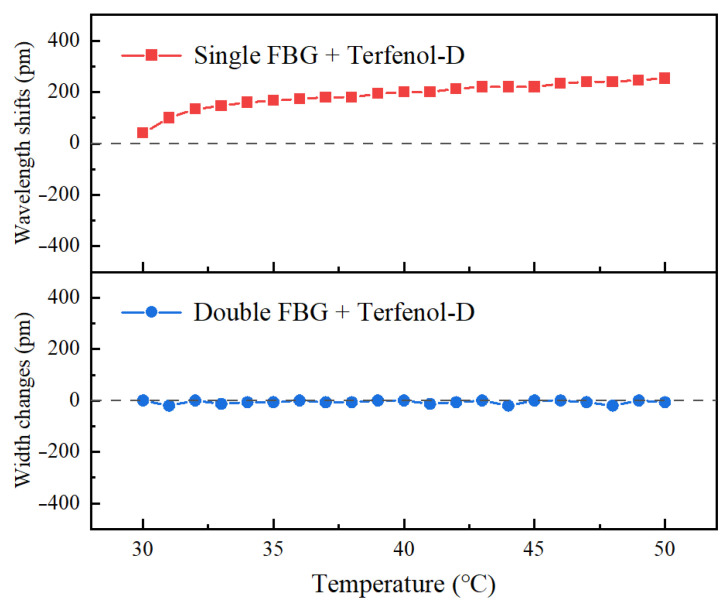
The average responses of the single FBG and Terfenol-D and double FBG and Terfenol-D during temperature changes.

**Table 1 nanomaterials-13-02109-t001:** Some property parameters of Terfenol-D at 25 °C.

Characteristic	Unit	Value (25 °C)
Coefficient of thermal expansion	um/mK	11.0
Resistivity	Ohm	6.0 × 10^−7^
Relative permeability		2–12
Magnetostriction	ppm	1620

**Table 2 nanomaterials-13-02109-t002:** Performance comparison of different sensors of the magnetic field based on FBG and Terfenol-D.

	Sensitivity to the Magnemagnetic Induction Intensitytic Field	References
Sensor based on FBG and magnetostrictive polymer composite (50% mass of powdered Terfenol-D)	2.75 pm/mT	[45]
Thin-slice fiber Bragg grating-giant magnetostrictive material sensor	1.089 pm/mT	[46]
Optical FBG Sensor using optical frequency-domain reflectometry	2.2 pm/mT	[47]
Sensor based on Single FBG + Terfenol-D	6.093 pm/mT	This work
Sensor based on Double FBG + Terfenol-D	7.382 pm/mT	This work

## Data Availability

All available data are contained within the article.

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
