# Peer review of "Temperature-Independent Sensor of the Magnetic Field Based on FBG and Terfenol-D"

_nanomaterials, 2023, doi:10.3390/nano13142109_

Round 1

Reviewer 1 Report

1. The direction of magnetostriction is not fully described. From fig. 4 it is not possible to determine the location of the main axis.

2. It is not described whether the residual magnetization was taken into account.

3. Temperature independence is not justified.

4. The temperature range of 30 - 50 degrees Celsius does not prove temperature independence. A small temperature spread and the effect of this may simply be unnoticed by the authors.

5. There is no description of the physics behind why the double fiber Bragg grating (FBG) with Terfenol-D is temperature independent.

Author Response

Thank you for your letter and for the reviewers’ comments concerning our manuscript entitled “Temperature-independent magnetic field sensor based on FBG and Terfenol-D” (nanomaterials-2497157). Those comments are valuable and helpful for revising and improving our paper, as well as the important guiding significance to our research. We have studied the comments carefully and have made a correction which we hope meet with approval. Revised portions are marked in red on the paper.

Reviewer 2 Report

This is a very well written text and on an interesting topic. I suggest publication after major revisions as following:

“magnetic field sensor” is ambiguous because can be interpreted also in the general sens of sensors based on magnetic field. I suggest to clarify it in title, abstract and all the manuscript by writing “sensor of the magnetic field”. (for instance figure 7 reports correctly the magnetic induction).

Also in the introduction it would be useful, although not indispensable, to clarify if the sensor measures directly the magnetic induction or the magnetizing field.

Add more information on the nanostructure of the terfenol, to fit better in the scope of the journal. For instance from the Rietveld of the xrd pattern one can infer on the crystalline size and comment on its relevance (size versus properties).

Figure 4: how was it possible the identification of the directions? Explain in full in text. It is not at all obivuous.

More characterization of the terfenol would be required (composition? Oxidation? Moisture resistance? Ageing due to oxidation and moisture?). These information, although not all requiring new experimental data, help fitting a journal about materials.

Author Response

(The authors gave the same response as above.)

Round 2

Reviewer 1 Report

Accept in present form

Reviewer 2 Report

the manuscript has been improved although more information on the nanostructure of the material would be useful. If this is not possible, the manuscript may still probably be published